# Calculation of an Average Vehicle’s Sideways Acceleration on Small Roundabouts

**DOI:** 10.3390/s22134978

**Published:** 2022-07-01

**Authors:** Juraj Jagelčák, Jozef Gnap, Mariusz Kostrzewski, Ondrej Kuba, Jaroslav Frnda

**Affiliations:** 1Department of Road and Urban Transport, Faculty of Operation and Economics of Transport and Communication, University of Zilina, 010 26 Zilina, Slovakia; juraj.jagelcak@uniza.sk (J.J.); kuba17@stud.uniza.sk (O.K.); 2Division for Construction and Operation of Means of Transport, Faculty of Transport, Warsaw University of Technology, St. Koszykowa 75, 00-662 Warsaw, Poland; mariusz.kostrzewski@pw.edu.pl; 3Department of Quantitative Methods and Economic Informatics, Faculty of Operation and Economics of Transport and Communications, University of Zilina, 010 26 Zilina, Slovakia; jaroslav.frnda@uniza.sk

**Keywords:** road safety, lateral acceleration, angular speed, roundabout, yaw, roll

## Abstract

The calculation of the average sideways acceleration, based on speed and angular velocity on small roundabouts for a vehicle of up to 3.5 t gross vehicle mass, is described in this paper. Calculations of the turning radius are derived from angular velocity and an automatic selection of events, based on the lateral acceleration of the coefficient of variation within a defined time window. The calculation of the turning radius based on speed and angular velocity yields almost identical results to the calculation of the turning radius by the three-point method using GPS coordinates, as described in previous research. This means that the calculation of the turning radius, derived from the speed of GNSS/INS dual-antenna sensor and gyroscope data, yields similar results to those from the computation of the turning radius derived from the coordinates of a GNSS/INS dual-antenna sensor. The research results can be used in the development of sensors to improve road safety.

## 1. Introduction

The stability of movement in vehicles with cargo in public traffic is one of the key aspects of ongoing research toward the improvement of road transport safety. This stability can be especially challenging when a vehicle negotiates curves or roundabouts. Therefore, the aim of this paper is to expand on our previous research [1] by studying long average accelerations (minimum duration equal to 1 s) with a focus on small roundabouts, and to present the possibilities of GNSS/INS sensor application in the computation of the turning radius based on speed and angular velocity measurements (noted from now as *R*3). The obtained values of turning radius *R*3 are compared with the turning radii *R*1 and *R*2 described in our previous publication [1]. A new method of the automatic selection of events, based on the coefficient of variation, is presented together with the division of roundabout turns into individual quadrants based on sensor yaw data.

## 2. Literature Review

According to [2], roundabouts have been used for years to reduce traffic hazards [3,4,5] and to ensure a proper capacity [6]. In the most frequently applied models of roundabout design, such as the Dutch (Dutch Information and Technology Platform, CROW model) and American models (FHWA, Federal Highway Administration model) [2], it is important that a vehicle’s turn through a roundabout consists of several interconnected radii. Turning radius, computed based on methods presented in relevant research, is developed with reference to the parameters of roundabout design. Moreover, the designed structure changes during its exploitation. Therefore, it is even more important to take current road conditions into account (in this context, especially roundabout condition), as in the method investigated in our paper.

The authors of [2] investigated maximum speed definition on single-lane roundabouts. They applied a precise GNSS (global movement navigation trajectory satellite in an urban system) device to obtain data during rides. The authors of [7] investigated and compared data obtained with GPS/GNSS and with stand-alone GPS. The driving maneuvers in their research were realized through a series of roundabouts. Many improvements in position accuracy were achieved using GPS/GNSS in comparison with GPS, especially for 10 Hz multi-GNSS. Meanwhile, the authors of [8] analyzed vehicle speed driving on an urban single-lane roundabout using both the abovementioned Dutch and American design models. It is important in roundabout design to ensure appropriate speed negotiation; therefore, the authors analyzed and compared the measured speed with the designed speed for the two mentioned models. Their research is ongoing.

A proper determination of parameters ensuring the safety of vehicles maneuvering on roundabouts, and at the same time, that of the passengers, requires the measurement of actual data collection. The authors of [9] proposed to collect such data, not solely within the roundabout, but also 100 m (including segmentation of this length) before entering and after exiting such road features. The authors of [10] developed a system for data collection on various types of infrastructure (this system requires a GPS receiver coupled with a three-axis accelerometer, a three-axis gyroscope, and a three-axis magnetometer). According to the authors, this elaborate system worked promisingly in a city, on a designated section of highway at a speed of 120 km/h, and on a roundabout.

Various authors have applied different methods to study roundabouts and the dynamic behaviors of vehicles driving through them. 

In previous years, as mentioned in paper [11], map-matching algorithms did not meet research expectations in the case of roundabouts, as precise identification of roads was difficult (this difficulty was caused by the fact that map-matching algorithms are highly dependent on road network characteristics). The authors applied fuzzy logic as a method, which may have improved results, since it deals with qualitative terms and linguistic vagueness together with human intervention.

The authors of [12] applied cellular automata for computer simulation to increase roundabout capacity by modification of road traffic rules. The authors analyzed the congestion of each lane on multilane roundabouts. Simulation methods were also used in the research of [13]. The authors presented the results of traffic simulation experiments, comparing solutions with regard to a small roundabout, a signal-controlled intersection, and a roundabout of the turbo type. The latter was characterized by significant improvement, in contrast to the former two scenarios.

Another method of gathering data to study roundabouts, the dynamic behavior of vehicles, and their analyses, is video-sequence processing, as developed by [14]. The authors developed an origin/destination matrix, compiled a vehicle classification system, and tracked individual vehicle trajectories together with certain data, such as corresponding speeds and acceleration along roads. The video-recording set was installed above a roundabout, a strategy that may involve limitations for research on the acceleration of certain vehicles, as well as for the research mentioned directly in the paper.

Road safety and its improvement through cargo securing was studied by the authors of [15]. In the study, authors performed eight rides in a truck and recorded data including acceleration. The main focus was on the value of the shock acceleration coefficients that were statistically evaluated. The authors also verified three hypotheses and results were presented.

Machine learning for the application of risk analysis to increase the support of driver assistance systems in roundabouts was of interest in the research of [16]. The authors focused on connected and autonomous vehicles driving on multilane roundabouts. Their results confirmed a strong linear relationship between the variations in time-to-collision values.

Machine learning was also applied in other research, since driving of autonomous vehicles on roundabouts may be treated as a critical operation. For example, the authors of [17] investigated a predictive model to estimate a vehicle’s speed and steering angle for such a critical operation. The authors of [18] developed a dedicated maneuver planning module, which focused on negotiation during the entering of a vehicle into a roundabout. To ensure this was possible, the authors implemented a synthetic environment within the module code in order to determine the interaction capabilities of a running vehicle. The timely and safe approach and join onto a roundabout were also an interest of [19]. Their solution was developed with the application of convolutional neural networks and machine learning.

The analyses connected to dynamic behaviors on roundabouts in the case of autonomous vehicles, which are not directly addressed in the research presented in the current paper (yet worth mentioning in the context of future research agendas), were also investigated in [20,21,22,23], to mention a few.

In summary, we found a cohort of topics that are of interest to researchers in the context of roundabout analyses. These are:Data gathering and collection [1,7,8,9,10,11,12,14,15],Increasing capacity and traffic flow [12,13],Optimization of a certain parameter [2,24],Safety aspects (all papers are connected to this topic, but it was especially significant in the studies referenced here) [1,2,8,12,13,14,15,16],Application of methods and tools (map-matching algorithms [11], machine learning [14,16,17,18,19], simulation [12], etc.).

In our previous paper [1], we found that the analysis of parameter measurements related to vehicles driving through roundabouts was neglected in the literature; at that time, in the cohort of investigated references, only [25] considered the topic of roundabouts. We continued the literature screening and investigated the cohort of topics presented above with the application of a panoramic literature review method. This involved data gathering and collection of measurements through the increase in capacity and improvement of traffic flow, various safety aspects, and the optimization of particular parameters. We also observed the application of various methods in research related to driving on roundabouts. Nevertheless, we did not find stability analyses with regard to driving a vehicle conveying cargo on a roundabout, a fortiori, using turning radii based on speed and angular velocity measurements. As research on roundabouts continues to be important, this study addresses a research gap regarding the stability of cargo-laden vehicles while driving on small roundabouts.

The definition of a small roundabout should be clarified before the main research topic is addressed. According to [8], small roundabouts are characterized by a maximum diameter of 35 m. The authors of [26] define mini-roundabouts as characterized by diameters between 13 and 25 m, while they consider compact single-lane roundabouts to be characterized by diameters between 26 and 40 m. There appears to be a lack of consensus on the dimensions of small roundabouts; therefore, we herein define small roundabouts as single-lane roundabouts with diameters of up to 40 m.

## 3. Materials and Methods

The main goals of this section are: to present the materials and methods applied to continue the previous research; to present the application of the GNSS/INS sensor in the turning radius calculation based on measured angular velocity (noted from now as *R*3); and to compare the values of turning radii *R*1 and *R*2 from previous research [1]. The following regulations and norms are also considered: [27,28,29,30,31,32]. The general framework of this research methodology is given in Figure 1.

### 3.1. Data Evaluation

The calculation of the turning radii *R*1 and *R*2 is described in previous research [1]. The GNSS/INS speed data v and gyroscope data gz from IMU are indispensable in stipulating turning radius *R*3. The original formula based on [33] was customized for the purpose of the research, as in Equation (1):(1)$$R3=\frac{v}{\left|gz1000\right|}\;[m]$$
where v is the vehicle speed and gz1000 is the average angular velocity during 1000 ms.

The parameter of gz1000 is calculated as an absolute value, because angular velocity assigns positive values (in the right curves) and negative values (in the left curves). The obtained value for radius *R*3 is then averaged for selected events and roundabout quadrants.

As mentioned in previous research [1], Equation (1) does not consider the inclination of the road (positive or negative) or the vehicle due to inertia forces. This influence is considered in Section 4 in a statistical assessment of the results based on actual field tests. The road inclination was difficult to measure with the tests performed, where total inclination (roll) was measured, and inclination of vehicle and road were almost indistinguishable.

The previous research [1] showed the difference between R1 and R2, and current research shows that R3 is equal to R2 for small roundabouts, so we can say that the inclination has a less significant impact on R3 than on R1.

### 3.2. Automatic Selection of Events Based on Coefficient of Variation of ay1000 (SEL3)

The automatic selection of events based on steady lateral accelerations ay1000 (SEL1) and on MSE of R1 and R2 (SEL2) was explained in previous research [1].

The selection’s SEL3 goal is to specify the events owing to the calculation of the coefficient of variation CV of ay1000 within a time window of 1 s, as follows:(2)$$CV=\frac{\sigma }{\left|\mu \right|}$$
where σ represents the moving standard deviation of ay1000 within a time window of 1 s, and μ represents the moving mean of ay1000 within a time window of 1 s.

The aim of this selection is to assign the event of main lateral acceleration acting in a roundabout (an example is given in Figure 2). For this purpose, the boundaries of the SEL3 event are represented by the first value (denoted as ‘start’ in Figure 2) and last value (denoted as ‘end’ in Figure 2), smaller than or equal to the value of CV of 0.04. This value is suitable for the selection of main lateral accelerations in small roundabouts for a given sensor. 

### 3.3. Automatic Identification of Roundabout Quadrants Based on Yaw

To study the different quadrants of a roundabout, the data from SEL3 turns were divided into quadrants based on the yaw data of the sensor. An example of this division is shown in Figure 3. This allowed the study of different quadrants of roundabouts in more detail, and also the comparison of vehicles and individual rides in individual roundabout quadrants. Only full quadrant data within the range of a 0.2–0.25 turn were evaluated from SEL3 turns.

The following test scenarios and vehicles were investigated in the field tests. Examples of vehicles are given in Figure 4. 

Two pallet units with a mass of 1 ton and a low center of gravity were loaded into the van vehicles (denoted in Table 1 as V10 and V12). Vehicle combination V8 was tested with a 400 kg single-axle trailer, and vehicle combination V9 was tested with a 700 kg single-axle trailer. The position of the sensor and selected parameters of vehicles are indicated in Figure 5 and Table 1. 

The sensor position was allocated to the vehicle’s longitudinal axis; specifically, on the roof of a passenger vehicle and directly under the roof of a van, with the antennas fitted on the roof. A lower sensor position corresponded to lower lateral acceleration values.

### 3.4. Roundabout Testing Scenarios

Five roundabout testing scenarios (further denoted as TSC) were deployed on 4 roundabouts in the tests. TSC1 and TSC2 were accomplished on the same roundabout. TSCs and the selected parameters of TSCs are given in Figure 6 and in Table 2. Each vehicle passed through each TSC 4 times.

The tests were carried out over 8 nights between 10:00 p.m. and 4:00 a.m. in order to avoid traffic as much as possible.

TSC1 was conducted on the largest roundabout in the study, with a minimum radius of 10.22 m and a maximum radius of 17.85 m. The roundabout was entered from the southwest direction and exited at exit 9 in the southeast direction. TSC2 was conducted on the same roundabout as TSC1. The roundabout was entered from the northeast direction and exited in the southeast direction via exit 10.

TSC3 took place on a roundabout with a minimum radius of 9.11 m and a maximum radius of 17.71 m. Dimensionally, this was a similar roundabout to that used in TSC1 and TSC2. The roundabout was entered from the east. It was then exited to the north via exit 11.

TSC4 measurements were taken at a roundabout with a minimum radius of 7.37 m and a maximum radius of 14.91 m. The roundabout was entered from the northwest and exited at exit 9 to the southeast. In terms of the crossing trajectory, TSC4 had a very similar crossing trajectory to TSC1, as confirmed by the same number of quadrants.

TSC5 was conducted on a roundabout with a minimum radius of 7.21 m and a maximum radius of 14.98 m. In terms of dimensions, TSC5 had similar dimensions to TSC4, being the smallest roundabout tested. The roundabout was entered from the northwest and exited at exit 10 to the southeast. TSC5 had a very similar crossing trajectory to TSC2, as indicated by the same number of quadrants. TSC3 also had a similar trajectory, where a different number of quadrants occurred due to a different exit distribution.

### 3.5. Statistical Investigation of Data

Based on the data shown in Table 2 (especially the roundabout radius parameter), we assigned testing scenarios to 2 groups (clusters). The first group contained data gathered from TSC1 to TSC3, and the second comprised TSC4 and TSC5 data.

Firstly, we had to confirm our assumption that there was no statistical difference between the scenarios belonging to each cluster. As a verification method, we chose the Friedman test (a non-parametric method that tests differences between several classes or categories, working with continuous dependent variables) [34]. Although we performed test scenarios using many vehicles, they all traveled the same testing routes (TSC1 to TSC5). Therefore, data obtained from sensors were considered as dependent variables (speed and lateral acceleration). The Friedman test (significance level of 0.05) proved that there was no statistical difference between the gathered data within each group/cluster. 

Secondly, we planned to analyze the impact of different vehicle types on data collected from sensors, as well on *R*1, *R*2, and *R*3 calculations. We set 2 assumptions, as follows:A1: There is a statistical difference between vehicle types within each data cluster.A2: If we test individual roundabout quadrants instead of the entire roundabout, we will obtain different results compared to A1.

Both hypotheses were verified by the Friedman test. Since this test allows analysis of mean ranks for input variables, post hoc analysis allowed us to identify vehicles with the highest/lowest values of certain driving dynamics. Due to this analysis, we were also able to ascertain whether the calculation of all 3 turning radii were independent of individual driving styles. The results delivered by the Friedman test indicated that driving dynamics had an impact on the sensor variables of speed and lateral acceleration (confirming A1). We identified significant statistical differences in the collected sensor data related to V6 and V7 (V6 gained the highest values of speed and lateral acceleration, while V7 reached the lowest values; see Table 1). For both data clusters, V1 and V11 oscillated close to the upper decision limit (relatively high speed and acceleration compared to the other vehicle types). 

A2 testing indicated similar (although not identical) results. Results obtained in the cases of vehicles V6 and V7 were identified as significantly different compared to the other vehicles, and in the cases of vehicles V1 and V11 were often close to the upper limit. However, we also recognized one novel case worth underlining. In the first testing group (TSC1–TSC3), only in quadrants 1 and 2 were results in the case of vehicle V8 close to the lower decision limit. This indicates that after V8 entered the roundabout, it passed the first 2 parts of the roundabout with a lower speed than other vehicles. This information reflects our assumption that the driver’s experience could partially impact the results, but the more important factor was the vehicle type used. Thus, we can state that in general, there was no difference between the results obtained from A1 and A2, respectively. An overview of the collected data can be found in Table 3 and Table 4.

On the other hand, according to the measured data, higher speed implied higher lateral acceleration. We did not find any evidence regarding the impact of driving dynamics on *R*1–*R*3 calculation, which means that the *R*3 Formula (1) can be used in real applications.

To compare measured and calculated parameters, we needed to evaluate the quality of created regression models. We chose the following metrics:Mean squared error (MSE): measures the quality of the prediction model based on Euclidean distance. The square root of MSE (RMSE) has the same units as an estimated variable [35].*R*2: the coefficient of determination. It measures the goodness of fit of a prediction model.RES95: corresponds to the 95th percentile of the absolute value of residuals (errors).

## 4. Results

In the current section, Figure 7 gives the results of SEL3 divided into quadrants for all the TSCs. We provide equations for turning radii and lateral acceleration based on statistical evaluation of the data (Figure 8, Figure 9, Figure 10, Figure 11 and Figure 12). 

The linear regression models designated for turning radii *R*1 and *R*2 are presented in Figure 8; those for turning radii *R*3 and *R*2 are presented in Figure 9; those for measured lateral acceleration ayM vs. calculated average lateral acceleration ayC1 are shown in Figure 10; those for ayM vs. ayC2 are shown in Figure 11; and those for maximum lateral acceleration ayMax vs. ayM are reported in Figure 12.

The linear coefficient multiplies the predictor values (x-axis), while coefficient b (also called bias or intercept) is the point where the function crosses the y-axis.

Residuals are calculated after running the regression model and are depicted on the graph located below the scatter plot. Residuals represent differences between the observed values and the estimated values (vertical lines). The residual plot allowed us to validate the model represented by the line of best fit. The good regression model is characterized by symmetric residual distribution, as well as a high density of points that are close to the origin and a low density of points that are away from the origin. As seen in Figure 8, Figure 9, Figure 10, Figure 11 and Figure 12, residuals crossed the red lines only in a few cases. Red lines (RES95) describe 95th percentiles for the residuals.

A detailed look at Figure 9 shows an example of a very strong relationship between the measured and predicted values. As both variables are expressed in the same range and the coefficient a is close to one, we could have removed the intercept, so that the regression model was significant.

The linear regression models from Figure 8, Figure 9, Figure 10, Figure 11 and Figure 12 give an equation expressed as follows:(3)$$R1c_{SEL3}=0.73465\cdot R2-1.36679\;[m]$$
(4)$$ayC1=\frac{v^{2}}{0.73465\cdot R2-1.36679}\;[g]$$
(5)$$R3c_{SEL3}=1.00394\cdot R2\;[m]$$

Equation (5) demonstrates that $R3c_{SEL3}$ is almost identical to *R*2. The RES95 value shows that 95% of all residuals are allocated below the absolute error of 0.17 m. Based on modification of Equation (1), Equation (6) for the calculation of ayC2 can be expressed as:(6)$$ayC2=\frac{v\cdot \left|gz1000\right|}{9.81}\;[g]$$

Equation (6) is based on speed and angular velocity and is not influenced by the turning radii.

Based on the results presented in Figure 8 and Figure 10, the following formula for the calculation of ayMc1 can be stipulated:(7)$$ayMc1=0.89759\cdot \frac{v^{2}}{0.73465\cdot R2-1.36679\;}+0.04551\;[g]$$

Based on Equation (6) and the model given in Figure 11, the following formula for the calculation of ayMc2 can be designated:(8)$$ayMc2=1.04143\cdot \frac{v\cdot \left|gz1000\right|}{9.81}+0.05295\;[g]$$

Based on the results shown in Figure 8, Figure 10 and Figure 11, Equation (9) for the calculation of ayMaxc1 was obtained:(9)$$ayMaxc1=0.99680\cdot \left(0.89759\cdot \frac{v^{2}}{0.73465\cdot R2-1.36679\;}+0.04551\right)+0.03236\;[g]$$
(10)$$ayMaxc1=0.89471\cdot \frac{v^{2}}{0.73465\cdot R2-1.36679}+0.07772\;[g]$$

Based on Equation (6) and the results given in Figure 11 and Figure 12, Equations (11) and (12) for the calculation of ayMaxc2 were obtained:(11)$$ayMaxc2=0.99680\cdot \left(1.04143\cdot \frac{v\cdot \left|gz1000\right|}{9.81}+0.05295\right)+0.03236\;[g]$$
(12)$$ayMaxc2=1.03809\cdot \frac{v\cdot \left|gz1000\right|}{9.81}+0.08514\;[g]$$

## 5. Discussion

The aim of the tests was to measure the long-term accelerations for regular road traffic driving in predefined vehicles (categories of vehicles: M1, N1, and O1) on four small roundabouts with five TSCs.

Based on the gathered data, we created a methodology able to calculate turning radius and lateral acceleration in a simplified manner, by using widely available and relatively cheap sensors (GPS with gyroscope and accelerometer). Testing scenarios included the types of small-diameter roundabouts that dominate modern cities. As shown in Table 3 and Table 4, the average speed and lateral acceleration of tested vehicles for each testing scenario (TSC) were different. As the proposed methodology had to be applicable in actual situations, we tested the influence of different vehicle types and individual driving dynamics on turning radius computations. Statistical investigation proved that our approach eradicated these impacts. This knowledge was a significant support for the *R*3 vs. *R*2 comparison and indicated that *R*3 could be completely substituted by *R*2, which we believe is one of the main benefits of this article.

The differentiated accelerations could only be measured during multiple passages through roundabouts at different speeds. The SEL3 selection, based on the coefficient of variation of ay1000, was applied for the selection of events. The data from SEL3 were divided into quadrants based on yaw sensor data to enable study of the different quadrants of the roundabouts.

The average speed and lateral acceleration of SEL3 quadrants for individual vehicles and TSCs are given in Table 3, and those for individual quadrants in Table 4.

## 6. Conclusions

In this research, we identified dynamic events of lateral acceleration in small roundabouts for vehicles with a GVM of up to 3.5 t, because it is possible to achieve high lateral acceleration in small roundabouts. We aimed to study long lateral acceleration of vehicles passing through small roundabouts to compare the turning radii calculated by three different methods. We also divided the data measured on roundabout turns into four quadrants, which allowed the study of different quadrants of roundabouts in greater detail, and also enabled comparison of vehicles and individual rides in individual roundabout quadrants.

Based on the correlation models of turning radii, we developed a model for the calculation of turning radius, average lateral acceleration, and maximum lateral acceleration for small roundabouts. The models of turning radii were valid for small roundabouts within a range between 9 and 15 m for SEL3 selection of events and quadrant identification. The RMSE of the R1 vs. the R2 model was 0.44 m. We also proved that the turning radius *R*3 obtained from speed and angular velocity was equal to that of the turning radius *R*2 obtained from GPS coordinates in small roundabouts, which confirms two methods of turning radii calculation with almost identical results. The RMSE of the ayM vs. the ayC1 model was 0.0165 g, while for ayM vs. ayC2, it was 0.0170 g, which indicates that ayC1 calculation gives slightly better results than ayC2. The models are more precise in comparison to previous research. The data for the model of calculation of turning radius ought to be periodically collected and analyzed, as we plan to develop a tool allowing feedback for the employees responsible for fleet management. However, this concept will be a matter of future research alongside other research agendas mentioned below.

Navigation devices are accessible to almost everyone, whether on a smartphone, built into a vehicle’s infotainment system, or as an external device. Truck drivers use them frequently, but they are also often used by car drivers to monitor traffic situations. The algorithm proposed for calculating accelerations could be incorporated into navigation applications. Every smartphone is equipped with sensors, including an accelerometer, a gyroscope, and a GPS sensor. Our algorithm ensures the possibility of determining the radius of each roundabout by collecting anonymous navigation data. Following data collection, the speed that should not be exceeded when crossing the roundabout would be calculated based on the collected data and on our model. The driver would be warned to slow the vehicle down in order to avoid skidding or overturning. This feature would be particularly useful in poor road conditions and would also help drivers with loads with a high center of gravity.

The main purpose of this paper was to produce results for application in freight-securing recommendations, in ensuring the stability of transported load units, and in calculating expected lateral accelerations through small roundabouts for a given speed and turning radius. The obtained results were found to be promising; therefore, in future research, new tests will be performed involving U-turns as well as on larger roundabouts, simulating scenarios involving cargo-securing decisions and the transport stability of load units. The results can be applied to improve road safety. When further research is considered, the development of a tool allowing feedback for the employees responsible for fleet management can become a significant future research agenda, both in the case of road safety and transport economics. Additionally, the influence on transport economics is related to the abovementioned fact of smartphones equipped with appropriate components (listed in the previous paragraph), and technologies which can support a driver during a ride. Our results can also be applied to algorithms used for autonomous road vehicles.

## Figures and Tables

**Figure 1 sensors-22-04978-f001:**
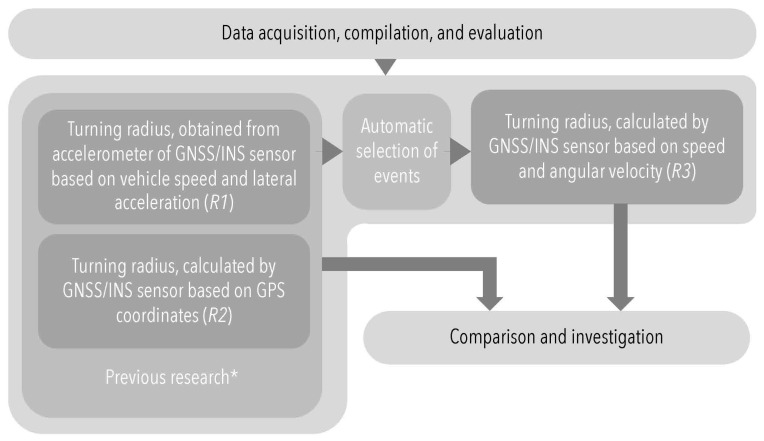
The research methodology (where * relates to [1]).

**Figure 2 sensors-22-04978-f002:**
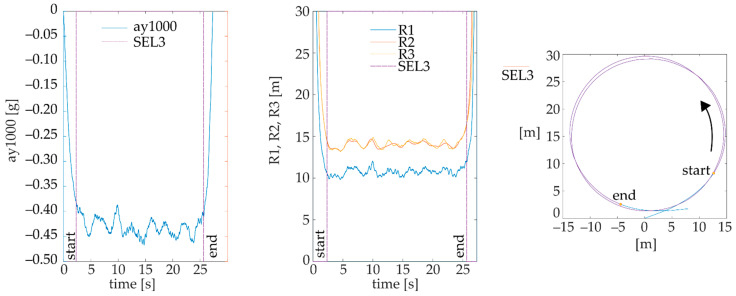
Example measurement for automatic selection of event SEL3 based on CV of ay1000.

**Figure 3 sensors-22-04978-f003:**
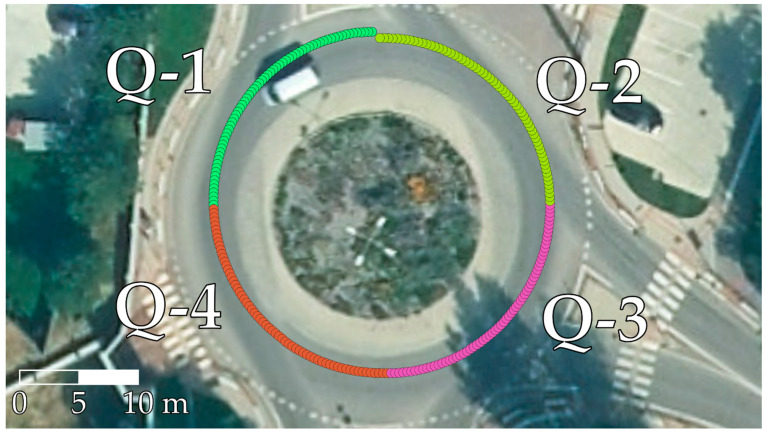
Example selection of quadrants denoted as Q-1 to Q-4 based on yaw, visualized on an orthophoto map layer of GKÚ Bratislava, NLC.

**Figure 4 sensors-22-04978-f004:**
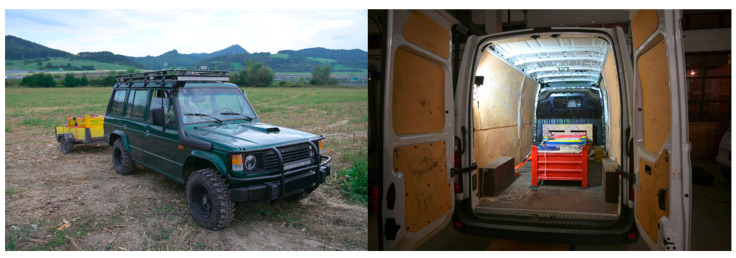
Vehicle combination V9 (in the **left** photo) and van vehicle V11 (in the **right** photo) applied in tests.

**Figure 5 sensors-22-04978-f005:**
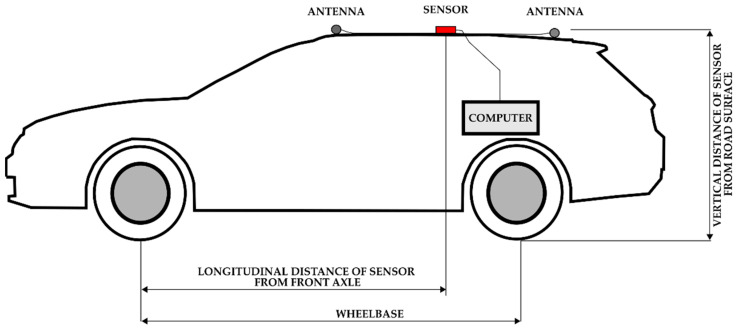
Scheme of test-setup installation.

**Figure 6 sensors-22-04978-f006:**
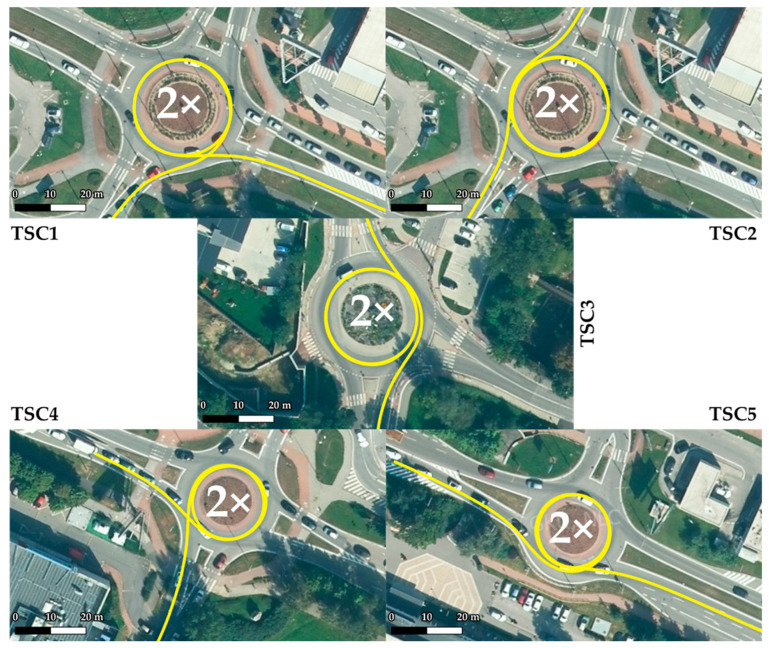
Roundabout testing scenarios 1–5 visualized on an orthophoto map layer of GKÚ Bratislava, NLC.

**Figure 7 sensors-22-04978-f007:**
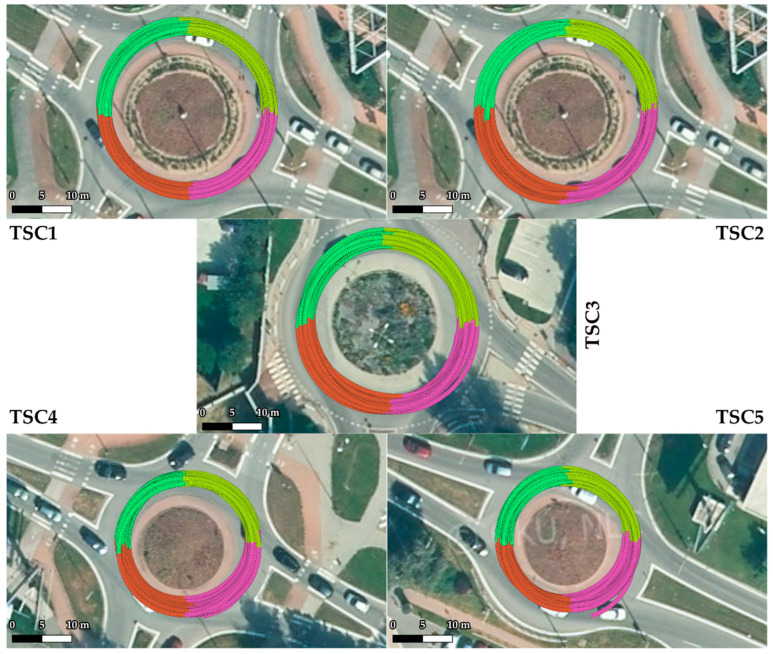
Results of SEL3 divided into quadrants Q-1 to Q-4 (colorized according to Figure 3) for roundabout TSC1–5, visualized on an orthophoto map layer of GKÚ Bratislava, NLC.

**Figure 8 sensors-22-04978-f008:**
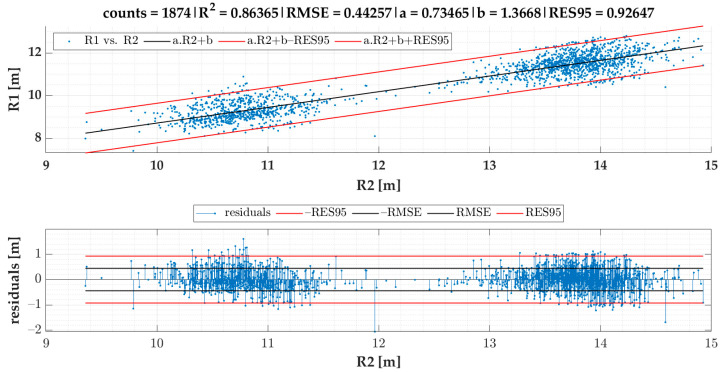
R1 vs. R2 in the case of events for SEL3 quadrants.

**Figure 9 sensors-22-04978-f009:**
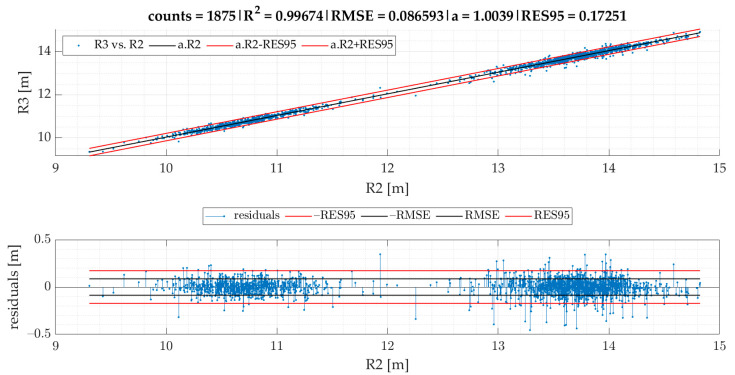
R3 vs. R2 in the case of events for SEL3 quadrants.

**Figure 10 sensors-22-04978-f010:**
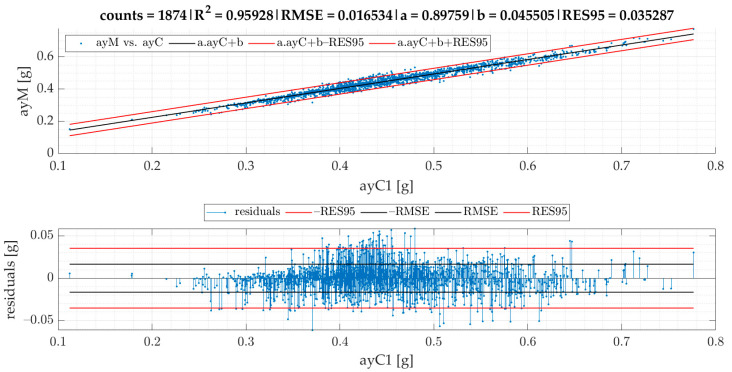
Calculated ayC1 vs. measured ayM of events for SEL3 quadrants.

**Figure 11 sensors-22-04978-f011:**
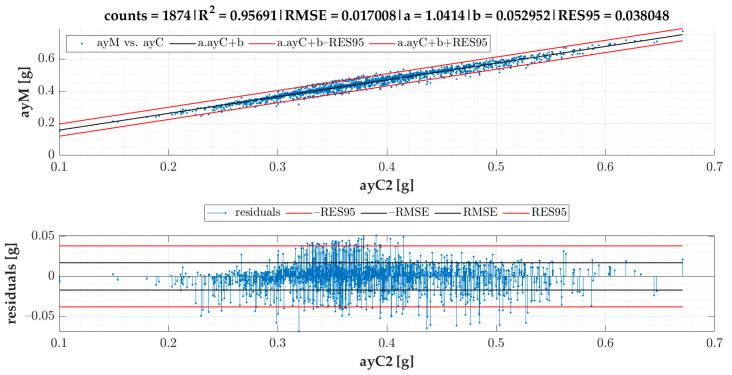
Calculated ayC2 vs. measured of events for SEL3 quadrants.

**Figure 12 sensors-22-04978-f012:**
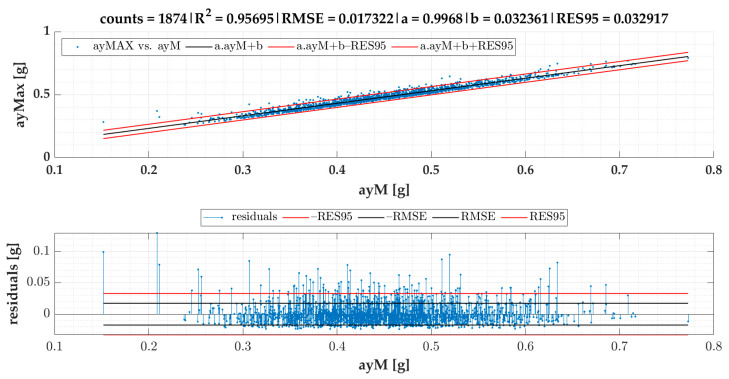
Maximum lateral acceleration ayMax vs. average lateral acceleration ayM of events for SEL3 quadrants.

**Table 1 sensors-22-04978-t001:** Selected parameters identifying the vehicles used in field tests.

ID	Vehicle Name	Manufacturing Year	Vehicle Category according to [30]	Vehicle Mass [kg]	Wheelbase [mm]	Longitudinal Distance of Sensor from Front Axle [mm]	Ratio of Position of the Sensor and Wheelbase [mm]	Vertical Distance of Sensor from Road Surface [mm]
V1	VW Golf	2001	M1	1265	2504	1805	0.72	1480
V2	VW Polo	2006	M1	1138	2441	1692	0.69	1480
V3	VW Polo	2004	M1	1033	2465	1721	0.7	1500
V4	VW Touareg	2003	M1G	2420	2865	1870	0.65	1713
V5	Opel Antara	2014	M1	1941	2710	1815	0.67	1705
V6	Škoda Fabia	2014	M1	1116	2460	1780	0.72	1550
V7	Honda Accord Combi	2008	M1	1766	2690	1850	0.69	1445
V8	VW Touareg; trailer	2003; 2005	M1G; O1	2850	2865; 2818	1870	0.65	1713
V9	Mitsubishi Pajero; trailer	1986; 1989	M1G; O1	2480	2690; 3350	1901	0.71	1890
V10	Renault Master	2019	N1	3350	4325	3020	0.7	2320
V11	Renault Master	2019	N1	2350	4325	3020	0.7	2320
V12	Renault Master	2014	N1	3330	4360	2920	0.67	2355
V13	Renault Master	2014	N1	2330	4360	2920	0.67	2355

**Table 2 sensors-22-04978-t002:** Selected parameters of TSC.

TSC	Minimum Radius of Roundabout [m]	Maximum Radius of Roundabout [m]	Minimum Radius *R*2 of Quadrant of SEL3 [m]	Maximum Radius *R*2 of Quadrant of SEL3 [m]	Number of Exits	Average Number of Turns from SEL3	Number of Quadrants
1	10.22	17.85	12.66	14.80	4	1.78	6
2	10.22	17.85	12.96	14.82	4	1.94	7
3	9.11	17.71	11.87	14.79	4	2.01	8
4	7.37	14.91	9.98	12.09	4	1.66	6
5	7.21	14.98	10.11	12.25	3	1.90	7

**Table 3 sensors-22-04978-t003:** Average speed v and lateral acceleration ay1000 of SEL3 quadrants for individual vehicles and TSC.

	TSC1		TSC2		TSC3		TSC4		TSC5		Total Average		Rank	
	*v*	*ay1000*	*v*	*ay1000*	*v*	*ay1000*	*v*	*ay1000*	*v*	*ay1000*	*v*	*ay1000*	*v*	*ay1000*
V1	26.50	−0.485	27.43	−0.510	26.28	−0.478	23.75	−0.485	25.73	−0.558	25.93	−0.503	3	3
V2	24.40	−0.382	25.67	−0.422	26.76	−0.461	24.53	−0.488	24.52	−0.475	25.16	−0.446	5	7
V3	23.96	−0.403	23.51	−0.389	23.86	−0.393	22.74	−0.461	20.73	−0.377	22.96	−0.403	11	11
V4	23.42	−0.391	24.33	−0.409	26.23	−0.480	23.13	−0.469	24.74	−0.538	24.43	−0.459	6	5
V5	24.44	−0.413	24.70	−0.413	24.83	−0.425	23.78	−0.477	22.38	−0.419	24.09	−0.429	8	9
V6	27.33	−0.504	28.83	−0.554	28.45	−0.545	25.78	−0.564	26.81	−0.606	27.52	−0.555	1	1
V7	21.04	−0.312	21.66	−0.336	22.68	−0.354	21.39	−0.392	20.92	−0.359	21.59	−0.350	13	13
V8	22.84	−0.369	23.75	−0.390	23.57	−0.397	21.76	−0.417	21.99	−0.417	22.83	−0.398	12	12
V9	24.80	−0.445	24.68	−0.448	24.99	−0.457	22.54	−0.467	22.85	−0.468	24.01	−0.457	9	6
V10	24.36	−0.419	25.60	−0.451	25.66	−0.449	22.10	−0.439	22.46	−0.435	24.12	−0.439	7	8
V11	26.02	−0.468	27.19	−0.497	27.41	−0.500	24.58	−0.522	24.98	−0.522	26.09	−0.503	2	2
V12	23.14	−0.374	24.04	−0.400	25.29	−0.437	22.23	−0.421	22.20	−0.420	23.45	−0.411	10	10
V13	24.73	−0.418	27.33	−0.499	28.41	−0.534	22.45	−0.435	24.41	−0.486	25.64	−0.479	4	4
Total average	24.37	−0.414	25.25	−0.438	25.67	−0.453	23.19	−0.466	23.41	−0.466	24.44	−0.448		
Rank	3	5	2	4	1	3	5	2	4	1				

**Table 4 sensors-22-04978-t004:** Average lateral acceleration ay1000 of SEL3 quadrants for individual vehicles, TSCs, and quadrants.

	V1	V2	V3	V4	V5	V6	V7	V8	V9	V10	V11	V12	V13	Total Average	Rank
TSC1	−0.485	−0.382	−0.403	−0.391	−0.413	−0.504	−0.312	−0.369	−0.445	−0.419	−0.468	−0.374	−0.418	−0.414	
Q-1	−0.465	−0.364	−0.388	−0.392	−0.398	−0.494	−0.304	−0.374	−0.441	−0.412	−0.459	−0.382	−0.422	−0.407	4
Q-2	−0.474	−0.380	−0.389	−0.392	−0.409	−0.511	−0.313	−0.355	−0.444	−0.413	−0.456	−0.364	−0.401	−0.408	3
Q-3	−0.518	−0.427	−0.437	−0.412	−0.447	−0.554	−0.324	−0.384	−0.452	−0.447	−0.507	−0.378	−0.419	−0.439	1
Q-4	−0.513	−0.376	−0.430	−0.374	−0.416	−0.458	−0.312	−0.371	−0.452	−0.416	−0.470	−0.375	−0.444	−0.413	2
TSC2	−0.510	−0.422	−0.389	−0.409	−0.413	−0.554	−0.336	−0.390	−0.448	−0.451	−0.497	−0.400	−0.499	−0.438	
Q-1	−0.510	−0.478	−0.385	−0.425	−0.438	−0.595	−0.318	−0.400	−0.447	−0.465	−0.518	−0.413	−0.534	−0.455	1
Q-2	−0.519	−0.442	−0.399	−0.422	−0.419	−0.577	−0.349	−0.404	−0.460	−0.472	−0.517	−0.394	−0.485	−0.449	2
Q-3	−0.510	−0.405	−0.386	−0.409	−0.407	−0.529	−0.338	−0.391	−0.455	−0.445	−0.495	−0.404	−0.493	−0.434	3
Q-4	−0.500	−0.392	−0.383	−0.388	−0.400	−0.536	−0.330	−0.370	−0.430	−0.430	−0.468	−0.397	−0.500	−0.424	4
TSC3	−0.478	−0.461	−0.393	−0.480	−0.425	−0.545	−0.354	−0.397	−0.457	−0.449	−0.500	−0.437	−0.534	−0.453	
Q-1	−0.459	−0.421	−0.385	−0.471	−0.405	−0.534	−0.360	−0.386	−0.448	−0.432	−0.483	−0.432	−0.518	−0.440	3
Q-2	−0.465	−0.427	−0.366	−0.435	−0.410	−0.510	−0.353	−0.366	−0.435	−0.431	−0.467	−0.424	−0.508	−0.429	4
Q-3	−0.492	−0.496	−0.395	−0.494	−0.443	−0.575	−0.340	−0.412	−0.450	−0.444	−0.509	−0.432	−0.535	−0.462	2
Q-4	−0.496	−0.494	−0.426	−0.514	−0.444	−0.560	−0.363	−0.421	−0.491	−0.485	−0.539	−0.456	−0.573	−0.480	1
TSC4	−0.485	−0.488	−0.461	−0.469	−0.477	−0.564	−0.392	−0.417	−0.467	−0.439	−0.522	−0.421	−0.435	−0.466	
Q-1	−0.480	−0.481	−0.457	−0.481	−0.477	−0.530	−0.388	−0.418	−0.468	−0.418	−0.510	−0.426	−0.438	−0.462	3
Q-22	−0.512	−0.494	−0.474	−0.489	−0.481	−0.563	−0.398	−0.430	−0.488	−0.456	−0.550	−0.426	−0.440	−0.478	2
Q-3	−0.462	−0.458	−0.435	−0.444	−0.461	−0.571	−0.378	−0.400	−0.450	−0.434	−0.501	−0.405	−0.420	−0.449	4
Q-4	−0.482	−0.542	−0.490	−0.465	−0.503	−0.585	−0.412	−0.426	−0.462	−0.433	−0.523	−0.436	−0.453	−0.480	1
TSC5	−0.558	−0.475	−0.377	−0.538	−0.419	−0.606	−0.359	−0.417	−0.468	−0.435	−0.522	−0.420	−0.486	−0.466	
Q-1	−0.551	−0.467	−0.352	−0.528	−0.399	−0.596	−0.326	−0.397	−0.452	−0.412	−0.509	−0.394	−0.461	−0.448	4
Q-2	−0.573	−0.476	−0.361	−0.538	−0.415	−0.583	−0.376	−0.429	−0.470	−0.445	−0.529	−0.440	−0.502	−0.471	2
Q-3	−0.543	−0.465	−0.403	−0.525	−0.420	−0.623	−0.380	−0.417	−0.478	−0.441	−0.510	−0.413	−0.477	−0.467	3
Q-4	−0.573	−0.512	−0.406	−0.585	−0.465	−0.638	−0.351	−0.435	−0.478	−0.446	−0.549	−0.443	−0.526	−0.493	1
Totalaverage	−0.503	−0.446	−0.403	−0.459	−0.429	−0.555	−0.350	−0.398	−0.457	−0.439	−0.503	−0.411	−0.479	−0.448

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
