# Peer review of "Calculation of an Average Vehicle’s Sideways Acceleration on Small Roundabouts"

_sensors, 2022, doi:10.3390/s22134978_

Round 1
Reviewer 1 Report
This paper developed a method for calculating average vehicle’s sideways acceleration in small roundabouts. The research is valuable.
1. The writing should be carefully checked. For example, Line 52, The driving maneuvers in their research were realized in between through a series of roundabouts. T
2. Describe more details about machine learning methods in the literature review.
3. What's the criteria for selecting different roundabout scenarios?
4. Using linear regression models may not represent the trend of Figure 11,12...
5. The major issue is that the paper lacks detailed description about the regression method.
Author Response
Revision Cover Letter - Calculation of average vehicle’s sideways acceleration in small roundabouts
Juraj Jagelčák, Jozef Gnap, Mariusz Kostrzewski, Ondrej Kuba, Jaroslav Frnda
Reviewer 1
Point 1
The writing should be carefully checked. For example, Line 52, The driving maneuvers in their research were realized in between through a series of roundabouts. T
Response 1
Thank you for this remark. You are perfectly right. It should be “The driving maneuvers in their research were realized in between through a series of roundabouts”, lines 51-52. We checked the language layer again.
Point 2
Describe more details about machine learning methods in the literature review.
Response 2
Machine learning is not applied in our research; however, to meet the suggestion of the honorable Reviewer we decided to expand the description of the literature review by the three following items: García Cuenca, L.; Sanchez-Soriano, J.; Puertas, E.; Fernandez Andrés, J.; Aliane, N. Machine Learning Techniques for Undertaking Roundabouts in Autonomous Driving. Sensors 2019, 19, 2386. https://doi.org/10.3390/s19102386; Capasso, A.; Bacchiani, G.; Molinari, D. Intelligent Roundabout Insertion using Deep Reinforcement Learning. In Proceedings of the 12th International Conference on Agents and Artificial Intelligence ICAART, Valletta, Malta, 22-24 February 2020; volume 2, pp. 378–385. https://doi.org/10.5220/0008915003780385; Wang, W.; Jiang, L.; Lin, S.l.; Fang, H.; Meng, Q. Imitation learning based decision-making for autonomous vehicle control at traffic roundabouts. Multimed. Tools. Appl. 2022. https://doi.org/10.1007/s11042-022-12300-9. Lines 99-108.
Point 3
What's the criteria for selecting different roundabout scenarios?
Response 3
We planned to differentiate the sizes of roundabouts and at the same time we decided to study urban roundabouts which typically are of rather small sizes.
Point 4
Using linear regression models may not represent the trend of Figure 11,12...
Response 4
High value of coefficient of determination allowed us to assume that linear regression models suit the best to results obtained in our research (estimation how a dependent variable changes as the independent variables’ change). Nevertheless, we realize that in subsequent case studies the equations may belong to nonlinear regression. Because we used simple linear regression with only two parameters, our models are robust to overfitting and can be easily implemented e.g. in software applications.
Point 5
The major issue is that the paper lacks detailed description about the regression method.
Response 5
Dear reviewer, thank you for your comment. We have added the following text to the section Results:
"The linear coefficient a multiplies the predictor values (x-axis) while coefficient b (also called bias or intercept) is the point where the function crosses the y-axis.
Residuals are calculated after running the regression model and are depicted on the graph located below the Scatter Plot. Residuals represent differences between the observed values and the estimated values (vertical lines). The residual plot allowed us to validate the model represented by the line of best fit. The good regression model is characterized by symmetric residual distribution as well as a high density of points that are close to the origin and a low density of points that are away from the origin. As it can be seen in Figure 8 to Figure 12, residuals crossed the red lines only in a few cases. Red lines (RES95) describe 95th percentiles for the residuals.
A detailed look at Figure 9 shows an example of a very strong relationship between the measured and predicted values. Because both variables are expressed in the same range and the coefficient a is close to 1, we could have removed the intercept, so the regression model is significant." Lines 315

Reviewer 2 Report
The paper calculate average sideways acceleration in small roundabouts. There is a application of modern sensors which can improve road safety based on experiment with various vehicles. Sophisticated sensors which can provide data very fast and cheap can be solution for some safety issues in road transport (in other modes of transport as well). It is very actual topic and new technologies provide still more accurate and better data for analysis. Mostly statistical analysis can be the essential tool to make practical output to improve road safety.
I have few remarks to the content:
- In statistics, it is customary to make a zero and an alternative hypothesis, zero hypothesis is normally about testing equality of parameters. E.g. applications for complementary topics: VLKOVSKY, Martin, NEUBAUER, Jiri, MALISEK, Jiri, MICHALEK, Jaroslav. Improvement of Road Safety through Appropriate Cargo Securing Using Outliers. SUSTAINABILITY, 2021, 13(5), 2688. ISSN 2071-1050. DOI: 10.3390/su13052688. It should be define the alternative hypothesis to find particular result (e.g. one parameter is higher than other one).
- The conclusion of the paper do not show specific measures in the case of finding unacceptable values for cargo securing. The use of sensors is just a tool, but how can road safety be increased? I believe, that the data can be periodically analyzed and make some feedback for the responsible person for fleet management. It can be applicable in big data as well.
- It would be useful to elaborate areas for further research in the context of key issues (road safety, transport economics, etc.).
Author Response
Revision Cover Letter - Calculation of average vehicle’s sideways acceleration in small roundabouts
Juraj Jagelčák, Jozef Gnap, Mariusz Kostrzewski, Ondrej Kuba, Jaroslav Frnda
Reviewer 2
Point 1
In statistics, it is customary to make a zero and an alternative hypothesis, zero hypothesis is normally about testing equality of parameters. E.g. applications for complementary topics: VLKOVSKY, Martin, NEUBAUER, Jiri, MALISEK, Jiri, MICHALEK, Jaroslav. Improvement of Road Safety through Appropriate Cargo Securing Using Outliers. SUSTAINABILITY, 2021, 13(5), 2688. ISSN 2071-1050. DOI: 10.3390/su13052688. It should be define the alternative hypothesis to find particular result (e.g. one parameter is higher than other one).
Response 1
Dear reviewer, thank you for your comment. In section 2.5, we set two hypotheses (statements) that we wanted to explore. Therefore H1 and H2 cannot be considered as zero and alternative hypothesis. Each of them (H1 and H2) has its zero and alternative hypothesis ( e.g. no statistical significance exists in a set of given observations) Because H1 and H2 was confusing, we changed them to Assumption 1 and Assumption 2, lines 264, 265, 266, 273, 278, 287.
The article Improvement of Road Safety through Appropriate Cargo Securing Using Outliers was read, and the reference was added to the text, lines 90-93.
Point 2
The conclusion of the paper do not show specific measures in the case of finding unacceptable values for cargo securing. The use of sensors is just a tool, but how can road safety be increased? I believe, that the data can be periodically analyzed and make some feedback for the responsible person for fleet management. It can be applicable in big data as well.
Response 2
Thank you for this suggestion. We added such an information in the conclusion of our paper: “The data for the model of calculation of turning radius ought to be periodically collected, analyzed as we plan to develop a tool allowing feedback for the employees responsible over fleet management. Such a concept however will be a matter of future research as other research agendas mentioned below.” Lines 402-405.
Point 3
It would be useful to elaborate areas for further research in the context of key issues (road safety, transport economics, etc.).
Response 3
Thank you for this remark. We have decided to add the following part to our paper: “When our further research agenda is considered, the development of a tool allowing feedback for the employees responsible over fleet management can become a significant future research agenda both in the case of road safety and transport economics. Additionally, the influence on transport economics is related to the abovementioned fact of smartphones equipped with appropriate components (listed in the previous paragraph) and technologies which can support a driver during a ride. Our results can also be applied in algorithms used for autonomous road vehicles.” Lines 424-431.

Reviewer 3 Report
The authors presenten an interesting work in the manuscript. I only some some minor comments.
1. More backgrounds should be added in the background.
2. The furture application of the proposed methods should be added.
3. Some of the sentences are awkard. A round of grammar check can imporve the manuscript.
Author Response
Revision Cover Letter - Calculation of average vehicle’s sideways acceleration in small roundabouts
Juraj Jagelčák, Jozef Gnap, Mariusz Kostrzewski, Ondrej Kuba, Jaroslav Frnda
Reviewer 3
Point 1
More backgrounds should be added in the background.
Response 1
At the beginning of our paper, we refer to a previously published article in which such background is indicated. The interested reader will refer to the previous paper in our opinion. We would not like to repeat information from the previous paper, because we would like to avoid being accused of self-plagiarism. We ask your understanding on this matter. Lines 29, 34.
Point 2
The future application of the proposed methods should be added.
Response 2
Thank you for this remark. We have decided to add the following part to our paper: “When our further research agenda is considered, the development of a tool allowing feedback for the employees responsible over fleet management can become a significant future research agenda both in the case of road safety and transport economics. Additionally, the influence on transport economics is related to the abovementioned fact of smartphones equipped with appropriate components (listed in the previous paragraph) and technologies which can support a driver during a ride. Our results can also be applied in algorithms used for autonomous road vehicles.” Lines 424-431.
Point 3
Some of the sentences are award. A round of grammar check can improve the manuscript.
Response 3
Thank you for this remark. You are perfectly right. We checked the language layer again.

Round 2
Reviewer 1 Report
No more comments.